# The variable source of the plasma sheet during a geomagnetic storm

L. M. Kistler [1,2] ✉, K. Asamura[3], S. Kasahara [4], Y. Miyoshi [2], C. G. Mouikis[1], K. Keika[4], S. M. Petrinec[5], M. L. Stevens [6], T. Hori[2], S. Yokota [7] & I. Shinohara [3]

Both solar wind and ionospheric sources contribute to the magnetotail plasma sheet, but how their contribution changes during a geomagnetic storm is an open question. The source is critical because the plasma sheet properties control the enhancement and decay rate of the ring current, the main cause of the geomagnetic field perturbations that define a geomagnetic storm. Here we use the solar wind composition to track the source and show that the plasma sheet source changes from predominantly solar wind to predominantly ionospheric as a storm develops. Additionally, we find that the ionospheric plasma during the storm main phase is initially dominated by singly ionized hydrogen ($H^+$), likely from the polar wind, a low energy outflow from the polar cap, and then transitions to the accelerated outflow from the dayside and nightside auroral regions, identified by singly ionized oxygen ($O^+$). These results reveal how the access to the magnetotail of the different sources can change quickly, impacting the storm development.

The magnetotail plasma sheet can contain plasma from both the solar wind and from the ionosphere. Figure 1 illustrates the main pathways for ions to reach the plasma sheet. When the interplanetary magnetic field (IMF) is southward, reconnection on the dayside creates open field lines that convect into the magnetotail, where they reconnect again[1]. Solar wind ions moving along these field lines can enter the plasma sheet. During northward IMF, the solar wind can enter either through double-lobe reconnection or along the flanks of the magnetosphere, in association with Kelvin-Helmholtz instabilities[2]. The ionospheric plasma also has access to the plasma sheet from several different regions. In the dayside cusp, energy from precipitating electrons and Poynting flux due to the interaction between the magnetosphere and the solar wind/IMF can heat and accelerate the ionospheric plasma, driving outflow[3]. During active times, the outflow has a significant $O^+$ component, with an $O^+/H^+$ ratio from 1 to 3[4]. Similarly, in the nightside auroral region, reconnection in the magnetotail also drives energy into the ionosphere, driving outflow that also has significant $O^+$[5]. Over the polar cap, a low-energy outflow called the polar wind is driven by an ambipolar electric field[6]. This outflow occurs continuously over the polar cap, for both northward and southward IMF, and consists mainly of $H^+$[7]. Because of its low energy, it is difficult to observe in the magnetosphere, but using a new measurement technique it has been shown that these cold ions are almost always present in the lobe region and likely constitute the majority of the ionospheric outflow[8].

Understanding how the access of ionospheric and solar wind sources to the magnetotail plasma sheet changes based on geomagnetic conditions is a key aspect of understanding magnetospheric dynamics during storms[9–12]. Geomagnetic storms occur when the IMF turns southward for an extended time. This both increases the amount of ionospheric plasma flowing into the magnetosphere and increases the convection electric field that transports ions to the nightside plasma sheet and into the inner magnetosphere. Due to the changes in the solar wind interaction with the magnetosphere, the contribution of the different plasma sources and source regions to the nightside plasma sheet can change. Recent simulations[12] have suggested that prior to the storm, the main source of the plasma sheet was the solar wind. But as the storm develops, the ionospheric

[1]University of New Hampshire, Durham, NH, USA. [2]Nagoya University, Nagoya, Japan. [3]Japan Aerospace Exploration Agency, Sagamihara, Japan. [4]University of Tokyo, Tokyo, Japan. [5]Lockheed Martin Advanced Technology Center, Palo Alto, CA, USA. [6]Harvard-Smithsonian Center for Astrophysics, Cambridge, MA, USA. [7]Osaka University, Toyonaka, Japan. ✉e-mail: Lynn.Kistler@unh.edu

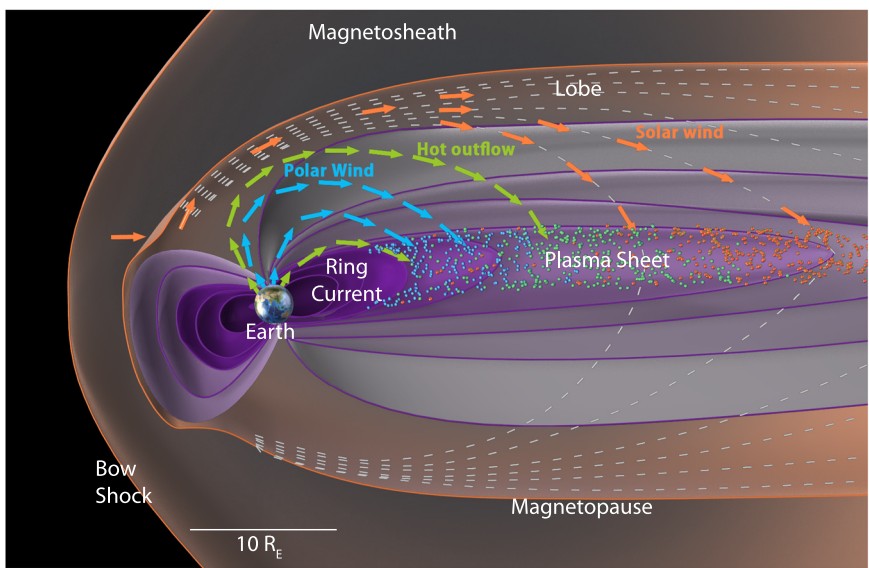

**Fig. 1 | Solar wind and ionospheric sources of the plasma sheet.** The pathways of the main sources of plasma to the plasma sheet during southward IMF are illustrated in a schematic of Earth's magnetosphere (based on Fig. 1 of ref. 1). The arrows represent the approximate paths of the different sources. Orange indicates the solar wind, coming in through the dayside magnetopause due to reconnection, and flowing along lobe field lines that convect to the plasma sheet. Green represents the hot outflow from the dayside (cusp) and nightside auroral regions. Blue represents the polar wind, outflowing from the polar cap. These sources mix in the plasma sheet with the contribution of the different sources changing with time.

plasma on open field lines is brought into the near-earth plasma sheet through reconnection, while the solar wind plasma enters further down the tail, leading to ionospheric plasma dominating the near-earth plasma sheet. The simulation also found a difference between the ionospheric $H^+$ from the polar wind and the ionospheric $O^+$ from the regions of more energetic outflow. The polar wind $H^+$ populated the lobe field lines earlier than the $O^+$ outflow, and so reached the plasma sheet first.

To determine observationally whether this source transition occurs requires a method to distinguish the sources in the plasma sheet. This is difficult because the main ion constituent, $H^+$, is contributed by both the solar wind and the ionosphere. However, the minor species of the two main sources are different. The solar wind heavy ions are highly charged, with $He^{++}$ and $O^{+6}$ the most abundant[13], while the ionosphere contributes mainly singly charged heavy ions, predominantly $O^+$ and $N^+$[14]. Thus, one way to determine the source of the plasma sheet plasma is to compare the plasma sheet composition with that of the solar wind. If the $He^{++}/H^+$ ratio is the same as in the solar wind, that indicates a predominantly solar wind source for the $H^+$. If the ratio is less than the solar wind, that implies there is a contribution of $H^+$ from the ionosphere. In addition, the cold polar cap polar wind source can be distinguished from the energetic outflow from the cusp or nightside aurora by the abundance of $O^+$. An $O^+$ rich source indicates that the source is one of the energetic outflow regions. A few studies have calculated the fraction of the plasma sheet that comes from the two sources statistically for quiet and active times using the composition[15,16], but did not examine the transition.

Recent work[17] did track the composition changes in the near-earth plasma sheet during a storm occurring in 1985 using the ion composition measurements from the Active Magnetospheric Particle Tracer Explorers (AMPTE)/Charge Energy Mass (CHEM) spectrometer[18]. However, there were no simultaneous solar wind data or interplanetary magnetic field (IMF) data available at that time with which to compare. They compared the plasma sheet composition with an average value for the solar wind $He^{++}/H^+$ ratio and found indications that the source changed from solar wind to ionospheric during the main phase of the storm. However, the $He^{++}/H^+$ ratio can be quite variable, changing by as much as a factor of 10 during storm times due to the enhanced heavy

ion abundance associated with coronal mass ejections (CME's)[19,20], and so tracking the source change quantitatively requires using simultaneously measured solar wind composition.

In this work, we use simultaneous measurements of the solar wind composition to track the source changes during one storm. We show that there are substantial changes in the composition of the near-earth plasma sheet as the storm develops. The source changes from solar wind-dominated to ionospheric-dominated at the start of the main phase. But for this storm, the ionospheric plasma is initially mainly $H^+$, indicating a polar wind source. Both the cusp and the nightside aurora outflow are significantly enhanced when a CME shock hits the earth during the main phase, and the hot $O^+$ outflow only reaches the nightside plasma sheet towards the end of the main phase. The results indicate that changes in the solar wind and IMF can cause significant changes to the near-earth plasma sheet composition, affecting the development of the ring current during a storm.

## Results
On Sept 7–8, 2017, a large geomagnetic storm with a minimum SYM-H of −150 nT occurred. This storm was part of a large space weather event, in which two X-class flares occurred and two interplanetary CME's impacted the earth. The event was the subject of a special issue in the journal Space Weather[21,22]. Here, we analyze the ion transport during this event using data from the magnetospheric multiscale (MMS) mission[23], the Japanese Arase mission[24], the Cluster mission[25], and the Wind mission[26]. The MMS mission and the Arase mission both have instruments that use a combination of an electrostatic analyzer followed by a time-of-flight measurement to separate $H^+$ from $He^{++}$ and also to identify the $O^+$ and $N^+$ group. The hot plasma composition analyzer (HPCA) on MMS[27] measures the ion composition over the energy range of about 1 eV to 40 keV/e. On Arase, the low-energy particle experiments−ion mass analyzer (LEPi)[28] measures ion composition over the range 0.01−25 keV/e and the medium energy particle experiments−ion mass analyzer (MEPi)[29] measures the ion composition over the energy range 10 keV/e to 180 keV/e. The Cluster/ Composition and Distribution Function (CODIF)[30] instrument uses a similar method to separate $H^+$ from the $O^+$ and $N^+$ group. Upstream of

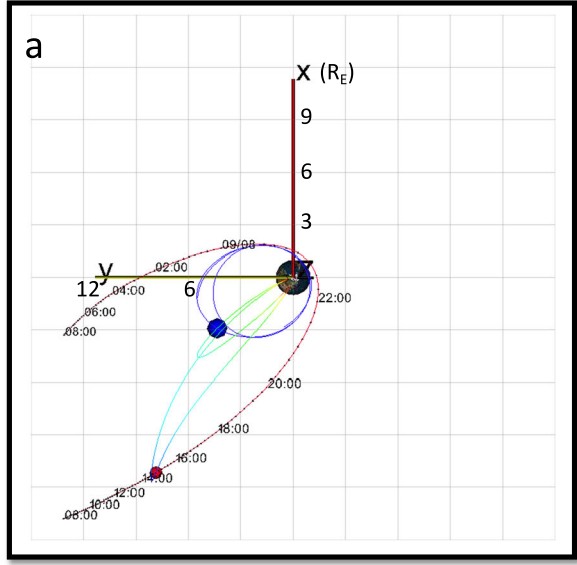

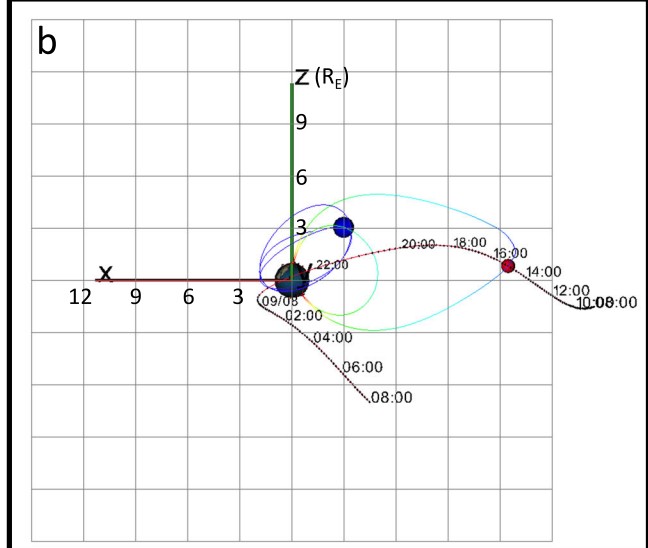

**Fig. 2 | Arase and MMS orbits.** Orbits of the Arase (purple) and MMS (Red) spacecraft from 8:00 on Sept 7, 2017 to 8:00 on Sept 8, 2017. **a** X-Y GSM projection and **b** X-Z GSM projection. The magnetic field lines that intersect the spacecraft (red and blue dots) at one location are shown with the rainbow-shaded lines. From 08:00–20:00 on Sept 7, MMS is moving inbound in the central plasma sheet from about $X = -14$ $R_E$ to $X = -6$ $R_E$, while Arase orbits in the inner magnetosphere, inside 6 $R_E$. From 1:00–8:00 UT on Sept 8, MMS moves duskward, towards the boundary layer and magnetosheath.

the magnetosphere, the solar wind experiment (SWE)[31] on the Wind spacecraft measures the solar wind $He^{++}/H^+$ ratio at the L1 Lagrangian point, see methods subsection WIND/SWE data.

Figure 2 shows the orbits of the Arase and MMS-1 satellites from before the storm through the storm's early recovery, projected into the X-Y and X-Z geocentric solar magnetospheric (GSM) planes. Arase spacecraft apogee was in the dusk-side near-earth plasma sheet. With an orbital period of 9.5 h, it samples the near-earth plasma sheet (defined as $L > 5$) for 6.5 h, then passes through the inner magnetosphere and comes out again to measure the near-earth plasma sheet for another 6.5 h. Thus, it samples the plasma sheet throughout the storm, with gaps for the inner magnetosphere passes. The MMS spacecraft has highly elliptical near-equatorial orbits with 25 Re apogee. Before the storm, MMS moves inbound through the central plasma sheet. The instruments turn off at 20:00 UT on Sept 7, before the spacecraft enters the inner magnetosphere. They turn on again at 1:00 UT on Sept 8, on their outbound pass, moving towards the dusk-side, and ultimately out into the magnetosheath.

The IMF conditions and the Arase data for the full-time period from 14:00 UT on Sept 6 to the end of Sept 8, covering both CME's, are shown in Fig. 3. Figure 3a–e show the total and Bz components of the IMF, the solar wind flow pressure, and the solar wind proton density and speed, all propagated to the location of Earth's bow shock, from the OMNI dataset[32]. Panel f shows the SYM-H index. Figure 3g–j shows the differential flux of $H^+$ and $O^+$ using the LEPi and MEPi data from Arase. Figure 3k gives the equatorial crossing of the magnetic field line that passes through the Arase spacecraft, calculated using a T89 field model. The times when the shocks from the two CMEs arrive at the bow shock are indicated with red vertical lines. The first shock, just before midnight on Sept 6, 2017, is associated with an increase in total magnetic field, solar wind proton density, and solar wind flow speed. The increased dynamic pressure drives an increase in SYM-H. Because there was no significant time period with southward Bz following the shock, this CME did not lead to a geomagnetic storm. The shock from the second CME arrived at 23:00 on Sept 7, 2017. This shock arrived after the IMF Bz had already turned southward, starting the main phase of the storm. The shock was associated with an increase in the total magnetic field, a strong southward Bz, a small increase in the proton density, and an increase in the solar wind flow speed. This shock drives

a strong storm. The four Arase orbits encompassing the geomagnetic storm are labeled in Fig. 3g.

Figure 4 shows data from 8:00 UT on Sept 7, 2017 to 21:00 on Sept 8. Figure 4a–c shows IMF Bz, the solar wind dynamic pressure, and the SYM-H index. The IMF Bz turns southward at 21:00 UT on Sept 7, a few hours before the arrival of the second CME. This leads to the drop in SYM-H (Fig. 4c), the signature of a storm main phase. When the CME hits at 23:00 UT, the magnetic field turns strongly southward. There is first a brief increase in SYM-H due to the increase in dynamic pressure, but then it decreases sharply, reaching its minimum of −150 nT on Sept 8 at 01:00 UT. There is a second southward turning during this event, beginning at about 11:30 UT on Sept 8. The SYM-H, which has been recovering, decreases again. Figures 4d–g show the proton and oxygen energy spectra from the MEPi and LEPi instruments on Arase, covering the hot plasma sheet energy range. The orbits are labeled in Fig. 4d, and separated by vertical lines. The first orbit shows strong $H^+$ and $O^+$ flux resulting from the first CME. The 2nd orbit occurs during the main phase of the storm. Arase is at the highest latitude during this orbit (Fig. 3k), on magnetic field lines close to the boundary layer that stretches further down the tail. This is evident by the low flux observed, with occasional very low flux time periods in $H^+$ that indicate excursions into the lobes. $O^+$ is also significantly reduced in this orbit, until the inbound pass. The 3rd orbit is again at lower latitudes and observes increased $H^+$ and $O^+$ fluxes. Towards the end of the 4th orbit, the second southward turning occurs, and the southward field continues into the final orbit. The next panel shows the $H^+$, $He^{++}$, and $O^+$ densities calculated over the energy range 1–200 keV/e, the hot plasma sheet energy. Only data in the near-earth plasma sheet outside $L = 5$ are included. During the main phase orbit, the second orbit in this plot, the densities when the spacecraft is outside the plasma sheet (in the lobe) are also excluded. Figure 4i–k shows the MMS/HPCA $H^+$ and $O^+$ energy spectra and the $H^+$, $He^{++}$, and $O^+$ densities. Finally, Fig. 4l compares the $He^{++}/H^+$ ratios measured at MMS and Arase with $He^{++}/H^+$ measured in the solar wind with Wind SWE. The $He^{++}$ measurement is difficult for these instruments, and there may be systematic errors on the order of a factor of 2. Thus our comparisons of the ratios between spacecraft are only considered significant if they are greater than this. The Wind spacecraft is located at the L1 point, and so there is a time delay between the measurement at L1 and the time the solar wind

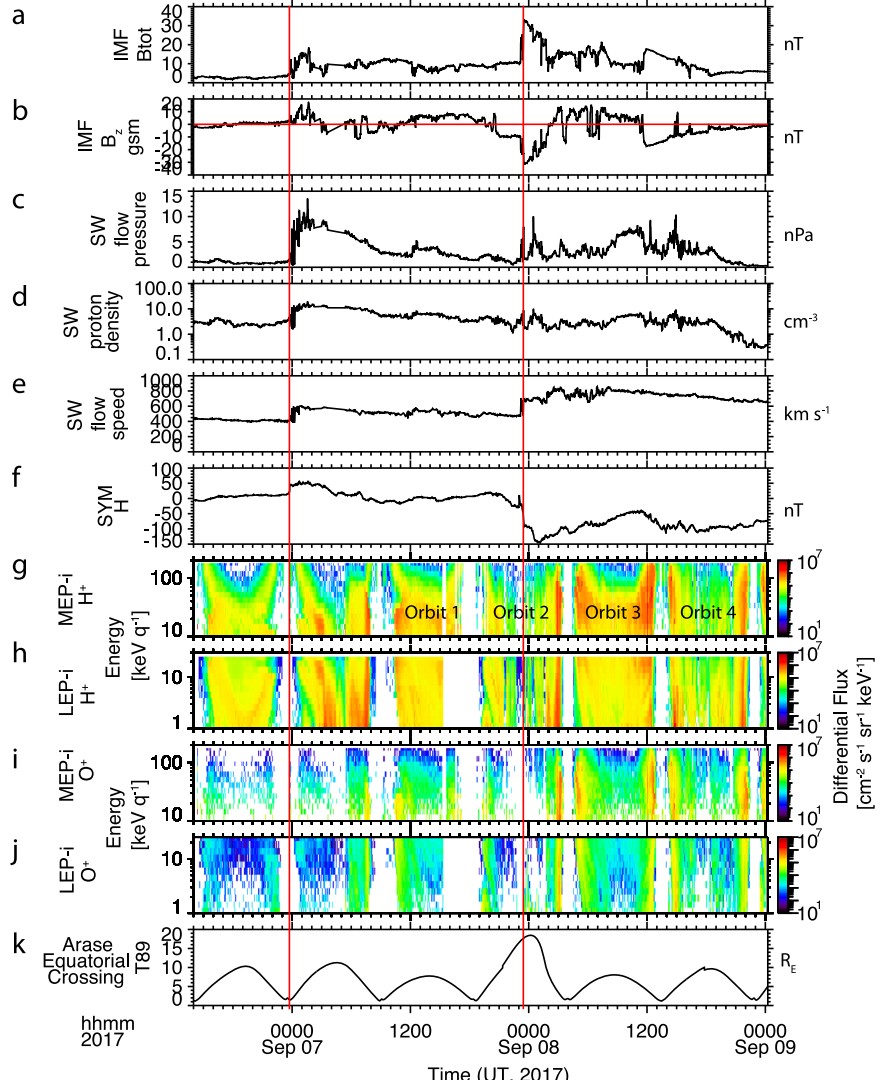

**Fig. 3 | Interplanetary driving conditions and Arase particle data for the full space weather event. a, b** Total and Bz component of the interplanetary magnetic field (IMF) from the OMNI dataset, propagated to the location of the Earth Bow Shock. **c–e** Solar Wind pressure, density and speed from the OMNI dataset propagated to the location of the Earth Bow Shock. **f** SYM-H index. **g, h** Differential energy flux for H⁺ from MEPi (10–180 keV) and LEPi (1–25 keV). **i, j** Differential energy flux for O⁺ from MEPi (10–180 keV/e) and LEPi (1–25 keV/e). **k** Equatorial crossing of the magnetic field line that passes through the Arase spacecraft, calculated using a T89 field model[33]. The red vertical lines indicate when the CMEs reach the nose of the bow shock. The red horizontal line in panel b indicates the zero line. The four orbits shown in Fig. 4 are labeled.

encounters the magnetosphere. The Wind data has been shifted by 45 min in this plot, to account for this travel time.

From Sept 07, 8:00–20:00 UT, before the storm, MMS is moving inbound in the central plasma sheet from 20 Re to 6.5 Re, as shown in Fig. 2. The H⁺ and He⁺⁺ densities increase as MMS moves inward. The O⁺ density is very low until 18:00 UT when it starts to increase. MMS is at about $L = 7$ at this time. During this inbound pass, the He⁺⁺/H⁺ ratio at MMS (red) agrees very well with the He⁺⁺/H⁺ ratio in the solar wind (black), indicating a solar wind source. At 20:00 UT, there is a sharp increase in the solar wind He⁺⁺/H⁺ ratio. Unfortunately, the HPCA instrument is turned off at this time. When HPCA turns back on at 1:00 UT on Sept 8, it measures a low He⁺⁺/H⁺ ratio. It also measures a high O⁺ density, about equal to H⁺, which persists as the spacecraft moves outward towards the dusk flank.

Starting at 4:00 UT on Sept 8th, a high flux population around 1 keV is observed in H⁺ (Fig. 4i), indicating that the MMS spacecraft has moved into the low-latitude boundary layer. It then moves into the magnetosheath at 9:00 UT. During this time, the MMS measurements and the Wind He⁺⁺/H⁺ ratios should agree because they both measure

solar wind origin plasma. As can be seen, there is excellent agreement between the two, with the significant changes in the solar wind He⁺⁺/H⁺ ratio observed on Wind clearly also seen in the MMS data. The good agreement validates the 45 min time shift that was used for the Wind data. The O⁺ density is low in the boundary layer. The HPCA O⁺ channel in the magnetosheath is background-dominated during this time, and so has been removed in the plot.

The second southward turning of the IMF occurs on Sept 8 at 12:00 UT. MMS is still in the magnetosheath, measuring the solar wind plasma. It exits the magnetosheath at around 19:15 UT, entering the more distant plasma sheet (about 20 Re). It measures a lower He⁺⁺/H⁺ ratio in the plasma sheet than that measured in the solar wind, again indicating an ionospheric contribution.

Now we consider the Arase He⁺⁺/H⁺ measurements. During the first orbit, when MMS is moving inbound before the storm, Arase measures a dense H⁺ plasma sheet with significant O⁺ as well, a result of the first CME (shown in Fig. 3). The He⁺⁺/H⁺ ratio is a factor of about 1.5 lower than the solar wind and MMS ratios but is still consistent with a strong contribution from a solar wind source. However, during the

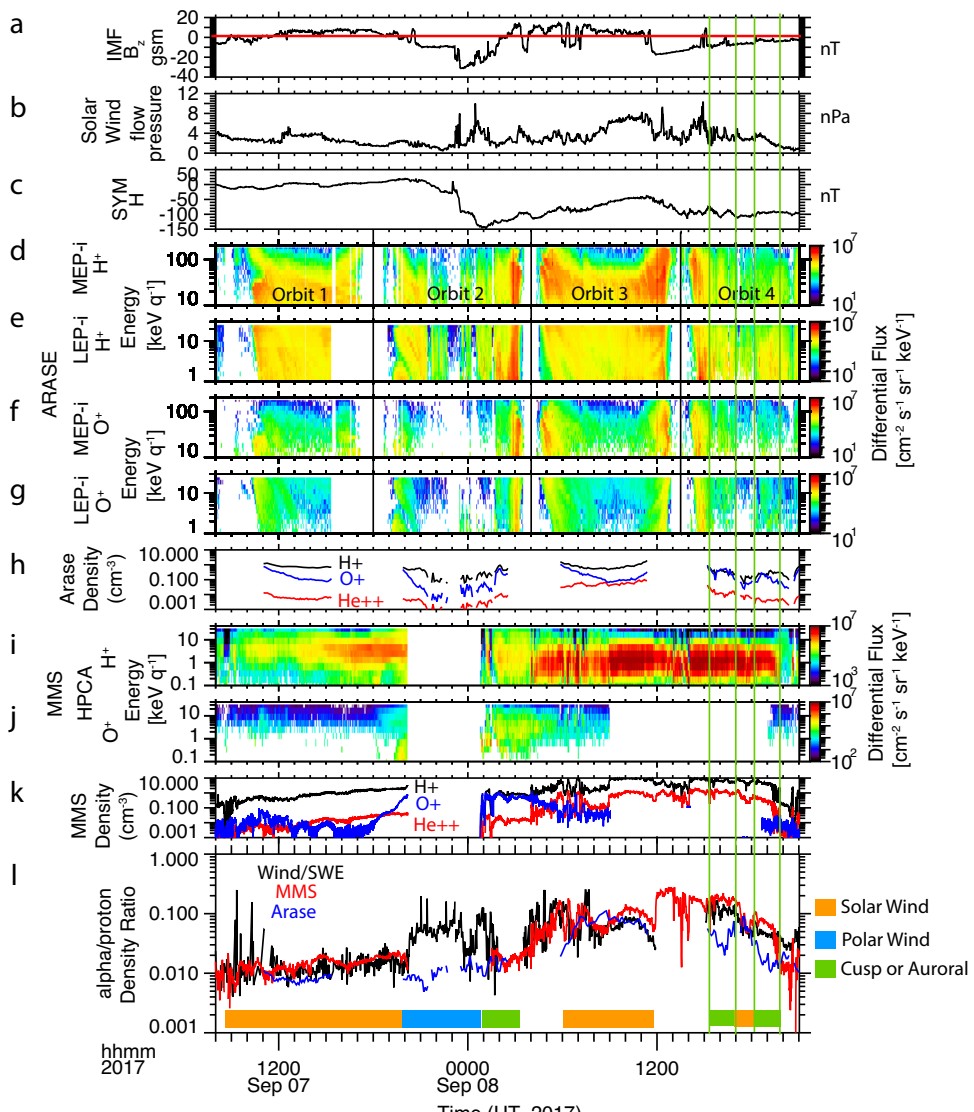

**Fig. 4 | Comparison of Arase and MMS measurements with solar wind composition. a**, **b** Total and Bz component of the interplanetary magnetic field (IMF) from the OMNI dataset, propagated to the location of the Earth Bow Shock. **c** SYM-H index, **d**–**g** Arase H+ and O+ Differential Flux, covering the range from 1–180 keV per q. **h** H+, O+, and He++ density at Arase integrated over the energy range 1–180 keV per q. **i**, **j** H+ and O+ differential flux at MMS. **k** H+, O+ and He++ Density at MMS. **l** He++/H+ density ratio, from Wind/SWE (black), MMS (red), and Arase (blue). The color bar at the bottom identifies the dominant source of the plasma in each location: solar wind (orange), polar wind (blue), and cusp or nightside auroral (green). The two cusp or auroral source time periods in orbit 4 are also indicated with green vertical lines.

main phase pass, the Arase He++/H+ ratio is a factor of 10 lower than the solar wind ratio, indicating a strong ionospheric contribution to the H+. The O+ density is quite low during most of this orbit. It finally increases on Sept 8 at about 1:50 UT. During the last two hours of this Arase pass, MMS/HPCA has turned back on, and the ratios measured at the two spacecraft are about the same. From 6:00 to 12:00 UT, when MMS is in the low-latitude boundary layer and magnetosheath, Arase has another pass outside $L = 5$. Now the Arase He++/H+ ratio in the near-earth plasma sheet is comparable to the solar wind measurements, showing the rather high He++/H+ values that are present in the solar wind. Thus, the plasma sheet has switched back to the mainly solar wind during this time. However, there is also energetic O+ present in the outer magnetosphere. The 4th Arase pass occurs from 15:00–20:00 UT, after the second southward turning. During this pass, Arase observes two time periods of about 1.5-hr duration when the He++/H+ ratio drops significantly below the solar wind value. These time periods, outlined with green vertical lines on the plot, are also accompanied by increases in O+ flux and density.

These MMS and Arase observations show that during the storm, the plasma sheet H+ source changes quite dramatically from a solar wind source to an ionospheric source and back again. When the IMF is southward, there are time periods when the He++/H+ ratio is significantly lower than in the solar wind, indicating an ionospheric source. When the IMF is northward, the He++/H+ ratio indicates the population is solar-wind-dominated.

The lack of O+ in the main phase orbit when the He++/H+ ratio is low, indicating that the H+ is mainly ionospheric, suggests that the source for the H+ during this time is the polar wind. During geomagnetic storms, energetic outflow from both the cusp and the nightside regions that are rich in O+ is often observed. However, it takes hours for the O+ to flow through the lobe to the plasma sheet. For O+ to enter the plasma sheet at the start of the main phase, it needs to already be present in the lobe prior to the main phase. In some storms[34,35], this has been observed. But in this case, this indicates that there was no cusp outflow close to the plasma sheet when the storm started.

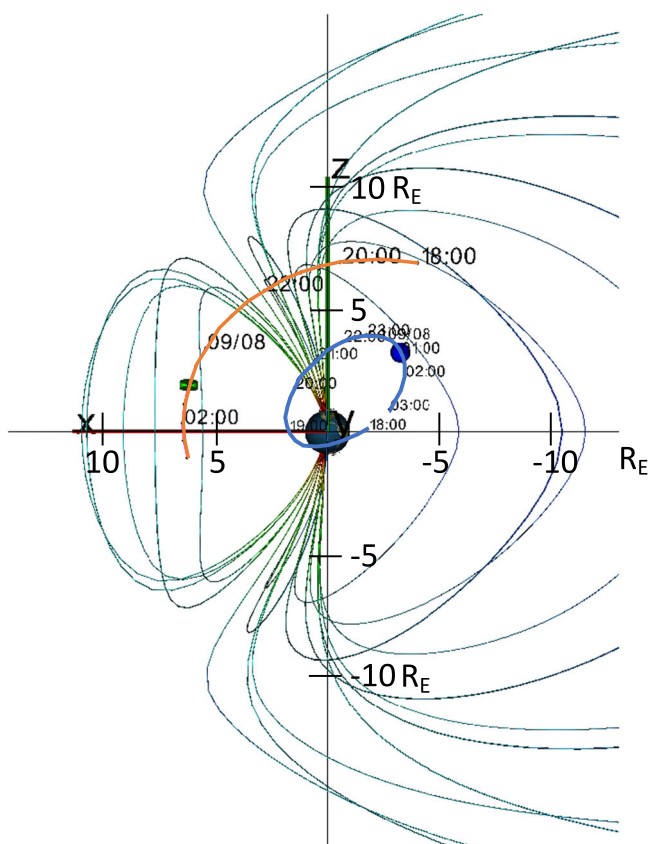

**Fig. 5 | Orbits of the Cluster and Arase spacecraft during the storm main phase.** Orbits of the Cluster (orange) and Arase (blue) spacecraft from 18:00 UT on Sept 7, 2017 to 3:00 UT on Sept 8, 2017 shown in an X-Z GSM projection with a model magnetospheric magnetic field.

To test this scenario requires measurements over the polar cap and in the lobes. There was a fortuitous pass of a Cluster satellite over the polar caps during the main phase of the storm. Figure 5 shows the Cluster orbit and the Arase orbit during the main phase pass. The cluster starts in the plasma sheet, moves through the lobe and over the polar cap, goes through the cusp itself, and then enters the inner magnetosphere. Figure 6a, b repeats the IMF Bz and SYM-H index. Figure 6c, e shows the Cluster $H^+$ and $O^+$ energy spectra from 40 eV to 40 keV using the CODIF instrument. At the start of the time period, when Cluster is in the plasma sheet prior to the southward turning of the IMF, there is very little $O^+$, consistent with the low $O^+$ observed at Arase a little later. But as Cluster moves over the polar cap from about 21:15-23:00 UT on Sept 7, it observes low-energy $H^+$ and $O^+$, a population characteristic of cusp-origin outflow that has convected tailward[36]. Figure 6d, f shows the pitch angle distributions of the low-energy $H^+$ and $O^+$, confirming that it is field-aligned and moving tailward, another characteristic of cusp-origin outflow. At 23:00 UT, when Cluster enters the cusp proper, intense energetic outflow is observed. The ion travel time to the plasma sheet depends on the magnetic field convection velocity as well as the ion energy. Even under high convection conditions, cusp $O^+$ outflow will take 1.5–2 h to reach the plasma sheet[37]. Most likely, the cusp outflow observed at Cluster started close to the time that IMF Bz turned southward at 20:30, and so it did not have time to reach the plasma sheet before the Arase measurements were made.

Another possible source for plasma sheet $O^+$ is the nightside aurora. Arase, in the near-earth plasma sheet, is well-placed to observe this outflow. Figure 6g, i shows the $H^+$ and $O^+$ energy spectra at Arase from 10 eV to 25 keV. Before 23:30 UT on Sept 7, there was

very little flux observed at low energies. Starting at 23:30 there is a bursty population observed in both $H^+$ and $O^+$ characteristic of nightside auroral outflow[34]. The pitch angle distributions of this bursty population, shown in Fig. 6h, j, confirm that it is upflowing. This outflow starts when the IMF-Bz becomes strongly negative, preceded by an increase in dynamic pressure. The enhanced reconnection driven by this strongly southward field apparently triggers reconnection on the nightside that sends energy via both Poynting flux and particle precipitation into the nightside auroral region, driving outflow. This strong outflow occurs three hours after the initial southward turning, and the observation of ionospheric $H^+$ in the hot plasma sheet.

Thus at 20:30 UT on Sept 7, when the IMF first turns southward, driving reconnection in the tail and inward convection that starts the storm main phase, there is no $O^+$ that has reached the plasma sheet. The only population available in the lobe to populate the plasma sheet is the ubiquitous polar wind population. So, in the early main phase of this storm, it is mainly $H^+$ that is being driven into the ring current. The $O^+$ from the cusp and nightside aurora are not observed in the plasma sheet until about 1:00–2:00 UT on Sept 8.

The color bar at the bottom of Fig. 4l shows a timeline of the changes to the plasma sheet that occurred during the storm. Prior to the storm, the plasma sheet $H^+$ comes predominantly from the solar wind. When the IMF turns southward, the plasma sheet source becomes ionospheric, but the composition is mainly $H^+$ from the polar wind. Cusp outflow starts around 20:30 UT on Sept 7, and nightside auroral outflow starts at 23:30. The $O^+$ from this energetic outflow is observed in the plasma sheet at about 1:00–2:00 UT on Sept 8, depending on the position. When the IMF Bz turns northward, there is another period of predominantly solar wind composition. And when IMF Bz turns south again there are periods when the source is again ionospheric.

## Discussion

In summary, there is a clear transition from the solar wind to the ionospheric source when the IMF turns southward and SYM-H decreases. The first ionospheric population to reach the plasma sheet is the $H^+$-rich cold polar wind. This is followed by the $O^+$-rich hot cusp or nightside auroral outflow source. Both the cusp and nightside auroral sources are active during, but not before, the main phase.

Whether the hot outflow enters the plasma sheet before the main phase or later in the storm depends on the solar wind drivers prior to the southward turning. Previous studies[34,35] have shown examples where enhanced dynamic pressure prior to the southward turning drove outflow so that the lobes and plasma sheet were already rich in $O^+$ at the start of the main phase when the enhanced convection started. This can lead to strong $O^+$ in the main phase ring current. In this storm, the cusp outflow did not start soon enough for the $O^+$ to reach the plasma sheet in time. The observations for this storm agree quite well with recent simulations[12]. The observations and simulation indicate that the time history of the solar wind drivers is important in determining when the different outflow sources reach the plasma sheet, and therefore how the composition of the storm-time ring current changes.

## Methods
### Arase data
The ERG Science Center (https://ergsc.isee.nagoya-u.ac.jp/index.shtml.en)[38] provides data for the LEPi and MEPi instruments as well as Arase orbit ephemeris data, and the spacecraft positions mapped using the T89 model[33], shown in Fig. 3k.

LEPi has two observational modes, normal mode and time-of-flight (TOF) mode. Only normal mode is used in this paper. In normal mode, LEPi provides a 3-D data product with 22.5-degree resolution in the azimuthal and spin phase directions for species that include $H^+$,

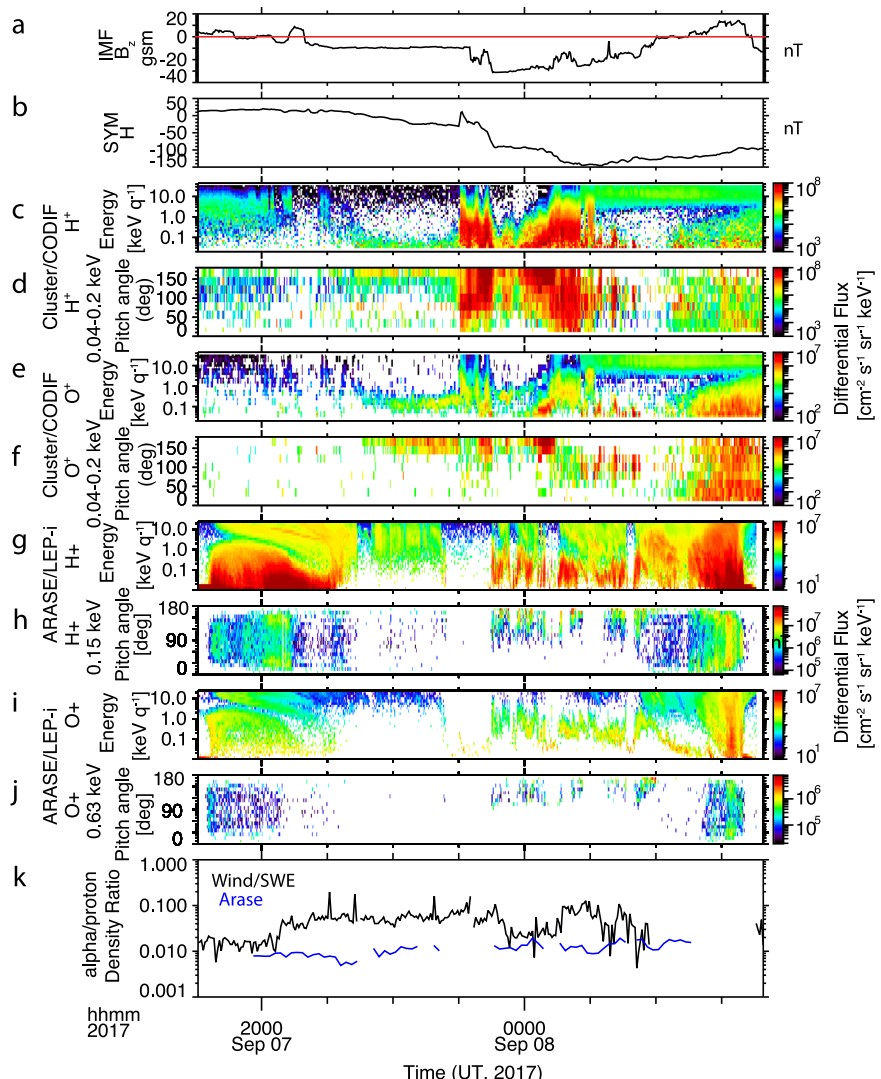

**Fig. 6 | Cusp and Nightside Aurora outflow sources observed by Cluster and Arase. a** Bz component of the interplanetary magnetic field (IMF), propagated to the location of the Earth Bow Shock during Arase orbit 2. **b** SYM-H index. **c**, **d** Cluster H⁺ differential flux spectrum and pitch angle distribution over the range 40–200 eV. **e**, **f** Cluster O⁺ differential flux spectrum and pitch angle distribution over the range 40–200 eV. **g**, **h** Arase H⁺ differential flux spectrum and pitch angle distribution at 150 eV. **i**, **j** Arase O⁺ differential flux spectrum and pitch angle distribution at 639 eV. **k** He⁺⁺/H⁺ density ratio at Wind/SWE (black) and at Arase (blue).

He⁺⁺, and O⁺ once per spin (8 s). In the second half of the first orbit in Fig. 3, LEPi is in TOF mode, so no data is shown and no density is determined.

For LEPi, the He⁺⁺ product is not provided to the public on a regular basis. The He⁺⁺ data for this event is available as a special dataset. The He⁺⁺ flux is determined using the same algorithms used for the other species. H⁺ and He⁺⁺ are well separated at least up to 10 keV[28], the upper limit for LEPi data used in the density calculation, so no special processing is needed.

MEPi similarly has both a Normal and a TOF mode, and both modes are used. In Normal mode, MEPi provides a 3D product (16 azimuthal angles, 16 spin phase angles, 16 energies) in one spin for 8 species. In TOF mode, the 16 azimuthal angles are summed into 4, and only one energy is measured in each spin phase. All 16 energy steps are sampled during one spin, one in each spin phase bin. For MEPi, H⁺, O⁺ and He⁺⁺ data are all publicly available

To calculate the plasma densities over the energy range 1–200 keV/q using these two instruments in different modes, we have used spin-averaged omnidirectional fluxes from each instrument and summed over the energy using LEPi from 1 to 10 keV/q, and MEPi from 10 to 200 keV/q using the following equation to calculate the partial density:

$$N = \sum_{i=a}^{i=b} 4\pi \sqrt{\frac{m}{2QE_i}} \Delta E_i J_i \tag{1}$$

where $E_i$ is the center value of the energy per charge step $i$, $\Delta E_i$ is the energy per charge range of the step, $m$ is the mass of the ion species, Q is the charge of the ion species, and $J_i$ is the differential number flux in $1/(cm^2 \, s \, ster \, \frac{keV}{e})$.

### MMS data
MMS data is available at the MMS Science Data Center, https://lasp.colorado.edu/mms/sdc/public/. The HPCA data used is taken in Fast Survey Mode. In this mode, a 3D product (8 azimuthal angles, 8 elevation angles, 16 energies) is provided every 10 s for the species H⁺, He⁺⁺, He⁺, and O⁺. The MMS differential flux and densities used are the standard L2 products. All MMS data is from spacecraft MMS-1.

### Cluster data
Cluster data is available at the Cluster Science Archive at https://csa.esac.int/csa-web/. For the CODIF instrument, a 3D data product

with 88 angles and either 16 or 31 energies, depending on the mode, is provided for $H^+$, $He^+$, and $O^+$. Standard differential flux products available at the Archive for the CODIF instrument are used. All data is from Cluster 4.

### WIND/SWE data

The standard Wind SWE proton and alpha particle spectra[39] are publicly available through https://cdaweb.gsfc.nasa.gov/, where they are archived by calendar date under the WI_SW-ION-DIST_SWE-FARADAY label.

A standard dataset that provides best-fit parameters to these particle spectra is available[40]. To estimate the ion species densities, temperatures, and velocities, species are modeled there with bi-Maxwellian phase space distribution functions, one for protons and one for alpha particles, and fit to the recorded spectra with a typical nonlinear least-squares regression. This is the standard model for low-energy solar wind ions at 1 AU, though certain deviations from the bi-Maxwellian peak shape may commonly be observed. The provided resource includes a number of quality flags to indicate periods where the fits are poor and the model may be an inappropriate functional form. The storm analyzed here is one such period.

However, in some cases, the bi-Maxwellian assumption is not valid. In particular, in some cases, the protons are double-peaked. This is one of the most commonly observed deviations from the standard model, and these spectra are readily modeled by including one additional Maxwellian proton population in the model regression. As in the standard treatment, the gyrotropy of the phase space distribution is maintained in the model by restricting relative flows between model populations to the local magnetic field direction. The tools to perform this fitting are available at https://github.com/JanusWind. The data using this method is included as a special dataset at https://doi.org/10.5281/zenodo.8313659. This treatment of the Wind SWE data has been previously employed in multiple studies[41-44].

## Data availability

The Arase MEPi data Level2 v01.03 (10.34515/DATA.ERG-03000) and LEPi data Level2 v01.00 (10.34515/DATA.ERG-05000), as well as the orbit data and mapped spacecraft position using the T89[33] magnetic field model are available at: https://ergsc.isee.nagoya-u.ac.jp/[38]. The MMS data are available at: https://lasp.colorado.edu/mms/sdc/public/ (files mms1_hpca_srvy_l2_ion_*_v4.1.0 and mms1_hpca_srvy_l2_moments_*_v4.1.0). Cluster data are available at https://csa.esac.esa.int/csa-web/ (files C4_CP_CIS_CODIF_HS_*_V180203.cdf). Special datasets created for this study, the $He^{++}$ LEPi dataset and the WIND/SWE data, are available at https://doi.org/10.5281/zenodo.8313659. The OMNI data is available from https://omniweb.gsfc.nasa.gov/. The data contained in Figs. 3, 4, and 6 are available at https://doi.org/10.5281/zenodo.8313659. The datasets generated during and/or analyzed during the current study are available from the corresponding author on request.

## Code availability

The software used to read and analyze the data is the publicly available SPEDAS software package[45] available at http://themis.ssl.berkeley.edu/software.shtml. The orbit plots were made using the Orbit Visualization Tool V3.O from https://ovt.irfu.se/. Tools to fit the Wind/SWE data are available at https://github.com/JanusWind. Code for the T89[33] magnetic field model is available at https://geo.phys.spbu.ru/~tsyganenko/empirical-models/magnetic_field/t89, and is also included as part of the SPEDAS software package.

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

## Acknowledgements

We acknowledge the dedicated efforts of the entire Arase, MMS, Wind and Cluster teams. We acknowledge the use of NASA/GSFC's Space Physics Data Facility's OMNIWeb service and OMNI data. The work at the University of New Hampshire is supported by NASA grants 80NSSC17K0643 and 80NSSC19K0073. L.M.K.'s work at Nagoya University was supported by ISEE, Nagoya University. K.A. is supported by JSPS, Grant-in-Aid for Science Research (21H04526, 22KK0046), Y.M. is supported by JSPS, Grant-in-Aid for Science Research (20H01959, 21H04526, 22KK0046, 22K21345, 23H01229). S.K., K.K., and S.Y. are supported by JSPS, Grant-in-Aid for Scientific Research (20H01957). S.P. is supported by NASA contracts 499935Q and 80NSSC18K1379.

## Author contributions

L.K. led the data analysis and writing of the paper. Y.M., K.A., S.K., K.K., S.Y., T.H., and I.S. developed the instruments for the Arase mission and processed and validated the data, making it available at the ERG Science Center. K.A. prepared the LEPi He++ special dataset. S.P. assisted in the analysis of the MMS/HPCA data. M.S. created the special Wind/SWE dataset and assisted in its interpretation. C.M. assisted in the analysis of the Cluster/CODIF data. All authors reviewed and edited the final manuscript.

## Competing interests

The authors declare no competing interests.
