## [Peer Review File · Nature Communications]

REVIEWER COMMENTS

Reviewer #1 (Remarks to the Author):

The major claim of this paper is the demonstration of the variability of the plasma sheet ion source during a geomagnetic storm. This is a very significant achievement, rendered possible by the fortunate availability of appropriate multiple spacecraft measurements at the right place at the right time, which is rarely the case.

I would suggest to slightly modify the title into

“The variable ion source of the plasma sheet during geomagnetic storms”

to emphasize the core result of the paper – which is not generally the plasma source, but the variability of the plasma source.

To be more concrete, the major point of interest in this paper is the demonstration of the change of the plasma sheet source from predominantly solar wind to predominantly ionospheric as the magnetic storm develops. Of course this does not imply that the same pattern should be expected for all magnetic storms, as the authors correctly point out at the end of the paper.

The methodology of the paper is appropriate and sound, with only a few weaknesses which can be easily remediated.

1. To fully substantiate the discussion, the authors have to provide quantitative information instead of the qualitative discussion in some critical parts of the paper. Please provide numbers at the following points:

- Lines 149-150: “the (Arase) He⁺⁺/H⁺ ratio is slightly lower than the solar wind” (the word “ratio” is missing here)

- Lines 151-152: “during the main phase pass, the Arase He⁺⁺/H⁺ ratio is significantly lower than the solar wind ratio” (it seems that the Arase He⁺⁺/H⁺ ratio is “significantly lower” even before the main phase starts – how do the authors interpret this?)

- Lines 154-155: “and the ratios measured at the two spacecraft are very consistent”.

- Line 156-158: "Now the Arase He⁺⁺/H⁺ ratio in the near-earth plasma sheet is consistent with the solar wind measurements, showing the rather high He⁺⁺/H⁺ values that are present in the solar wind" (consistent with = comparable to?)

2. Please include in the plot (Figs. 3 and 4) the error bars of the ratios (or at least address them in the text).

3. Lines 30-32: "The source of the hot (~1-50 keV) plasma sheet is important because its properties control the enhancement and decay rate of the ring current, the main signature of a geomagnetic storm" – the ring current is not a signature of a geomagnetic storm. A geomagnetic storm occurs because of the ring current intensification, hence the ring current is the cause of the storm, not its effect/signature. I would suggest the wording "... the main physical component (or agent) of a geomagnetic storm".

4. Please mark/indicate on Extended Data Fig 1 the start/end of the two CMEs mentioned, provide the exact times in the text and discuss the features of their identification.

5. Please add the date whenever UT is mentioned – at least the first time UT is mentioned in a paragraph (e.g., Line 155).

6. In Figure 3 the orbits of Arase should be indicated by e.g. vertical lines; this is important because of statements like this: "The O⁺ density is quite low during most of this orbit" (Line 153). I am suggesting this for the sake of the reader – he should be able to locate the described feature relatively easily.

7. Line 23: "because the solar wind He⁺⁺/H⁺ ratio can vary by as much as a factor of 10 during a storm" – please provide a comment/explanation and one or more references here.

8. Lines 161-163: "Arase observes two ~1.5-hr time periods when the He⁺⁺/H⁺ ratio drops significantly below the solar wind value. These time periods are also accompanied by increases in O⁺ flux and density" – O⁺ flux and density seem to decrease rather than increase. Please provide comment/explanation.

9. Lines 201-203: “The enhanced reconnection driven by this strongly southward field apparently triggers dynamic behavior on the nightside that drives the outflow” – please provide some suggestion or hypothesis on the “triggering of dynamic behavior” (both with regard to “triggering” and with regard to the “dynamic behavior”).

Reviewer #2 (Remarks to the Author):

This is a review of the paper “The source of the magnetotail plasma during a geomagnetic storm” submitted by Kistler et al. This paper looks at a fundamental question in magnetospheric science, namely what is the origin of near-Earth plasma. As the paper points out, there are two sources of magnetospheric plasma: the solar wind and the ionosphere. O⁺ clearly comes from the ionosphere but H⁺ can come from either source. This paper examines the source of plasma in the plasmashet using a combination of Arase, MMS, Cluster, WIND/SWE, and Cluster data. They also overcome the ambiguity regarding the source of H⁺ by using simultaneous monitoring of the He⁺⁺/H⁺ ratio in the solar wind and the magnetosphere. It has been shown in past work that this ratio could be used to determine how much of the H⁺ is of ionospheric or solar wind origin (although simultaneous monitoring was not available in the past work). Throughout the course of an event, the authors successfully track the composition change and the origin of the plasma.

Overall, this is an excellent and well written study well deserving of publication. It addresses an issue of fundamental importance to magnetospheric science (the origin of near-Earth plasma). It is important for space weather as magnetospheric composition affects the decay of the ring current. This topic also relates to the issue of ionospheric outflow which has broad applicability to planetary science and atmospheric evolution. This study employs excellent data analysis to convincingly show how H⁺ of ionospheric origin can dominate the storm main phase before the O⁺ rich outflow arrives. As H⁺ has often been neglected as an ionospheric source of plasma in comparison to O⁺ of less ambiguous origin, this is a compelling result. These results are also consistent with some recent simulations as the authors point out. It is my recommendation that this work be published as is.

Reviewer #3 (Remarks to the Author):

By combining data from multiple spacecraft, the authors present a convincing story on the time evolution and origins of different ion species in the plasma sheet during a geomagnetic storm.

Some earlier studies have considered the ion composition in the plasma sheet during storms, but the novelty of this study is in the simultaneous data from different locations in the magnetosphere – in the near-Earth (Arase) and far (MMS) plasma sheet, as well as crossing the lobes and polar cap (Cluster), and also upstream in the solar wind (Wind). By comparing the data from different ion species at different locations, they are able to give a clear global description of the ion composition at different times during the storm. I found that the paper is well written, and the conclusions are well supported by the data presented. The data has been made available, so that others are able to reproduce the study.

Some minor suggestions –

1. There are some older references that are relevant to this work:

Gloeckler, G., & Hamilton, D. C. (1987). AMPTE ion composition results. *Physica Scripta*, 1987(T18), 73.

Shelley, E. G. (1986). Magnetospheric energetic ions from the Earth's ionosphere. *Advances in space research*, 6(3), 121-132.

2. Since there is a lot of free space in Fig. 1, I would suggest adding more labels for the different regions of the magnetosphere to help to make the article more accessible to a general audience.

Response to the referees:

Thank you to the referees for the comments and suggestions. We have addressed all comments, as detailed below (responses in red). In addition, we had to make some changes to conform to the format for Nature Communications. The abstract needed to be shorter and could not include references. There needed to be headers, and there needed to be a paragraph at the end of the introduction that gives a summary of the major results and conclusions. These changes, and some associated changes required to put some of the information dropped from the abstract in the main text were also added.

Reviewer #1 (Remarks to the Author):

The major claim of this paper is the demonstration of the variability of the plasma sheet ion source during a geomagnetic storm. This is a very significant achievement, rendered possible by the fortunate availability of appropriate multiple spacecraft measurements at the right place at the right time, which is rarely the case.

I would suggest to slightly modify the title into
“The variable ion source of the plasma sheet during geomagnetic storms”
to emphasize the core result of the paper – which is not generally the plasma source, but the variability of the plasma source.

The title has been changed to “The variable source of the plasma sheet during a geomagnetic storm”

To be more concrete, the major point of interest in this paper is the demonstration of the change of the plasma sheet source from predominantly solar wind to predominantly ionospheric as the magnetic storm develops. Of course this does not imply that the same pattern should be expected for all magnetic storms, as the authors correctly point out at the end of the paper.

The methodology of the paper is appropriate and sound, with only a few weaknesses which can be easily remediated.

1. To fully substantiate the discussion, the authors have to provide quantitative information instead of the qualitative discussion in some critical parts of the paper. Please provide numbers at the following points:

- Lines 149-150: “the (Arase) $\text{He}^{++}/\text{H}^+$ ratio is slightly lower than the solar wind” (the word “ratio” is missing here)

- Lines 151-152: “during the main phase pass, the Arase $\text{He}^{++}/\text{H}^+$ ratio is significantly lower than the solar wind ratio” (it seems that the Arase $\text{He}^{++}/\text{H}^+$ ratio is “significantly lower” even before the main phase starts – how do the authors interpret this?)

- Lines 154-155: “and the ratios measured at the two spacecraft are very consistent”.

- Line 156-158: “Now the Arase $\text{He}^{++}/\text{H}^+$ ratio in the near-earth plasma sheet is consistent with the solar wind measurements, showing the rather high $\text{He}^{++}/\text{H}^+$ values that are present in the solar wind” (consistent with = comparable to?)

Have changed the wording of these to be more quantitative. (lines 208-224 tracked changes version)

2. Please include in the plot (Figs. 3 and 4) the error bars of the ratios (or at least address them in the text).

The statistical errors can be judged by the short term variations in the densities, and are small (<10%) for H⁺ and He⁺⁺ for most of this time period. The systematic errors are on the order of a factor of 2. For example, there are 4 MMS spacecraft very closely spaced so they should be measuring the same population, and they show differences on this order. A quantitative calibration of the alpha measurement is difficult to determine for all these instruments, and so we do not attribute any significance to the small differences on this order that are observed throughout. Our main arguments rest on differences of factors of 5-10 that are observed between instruments and similarities and differences in the temporal profiles. We have the sentence: “The He⁺⁺ measurement is difficult for these instruments, and there may be systematic errors on the order of a factor of 2. Thus our comparisons of the ratios between spacecraft are only considered significant if they are greater than this.” (lines 176-179 of “tracked changes” version)

3. Lines 30-32: “The source of the hot (~1-50 keV) plasma sheet is important because its properties control the enhancement and decay rate of the ring current, the main signature of a geomagnetic storm” – the ring current is not a signature of a geomagnetic storm. A geomagnetic storm occurs because of the ring current intensification, hence the ring current is the cause of the storm, not its effect/signature. I would suggest the wording “... the main physical component (or agent) of a geomagnetic storm”.

Changed to “The source is critical because the plasma sheet properties control the enhancement and decay rate of the ring current, the main cause of the geomagnetic field perturbations that define a geomagnetic storm.” (Now the second sentence of the abstract)

4. Please mark/indicate on Extended Data Fig 1 the start/end of the two CMEs mentioned, provide the exact times in the text and discuss the features of their identification.

The two CME’s are now marked in the figure, and the figure caption discusses the features. The orbits discussed in the main paper are also now labeled.

5. Please add the date whenever UT is mentioned – at least the first time UT is mentioned in a paragraph (e.g., Line 155).

Added dates (and also UT’s) in a number of places to make it clearer.

6. In Figure 3 the orbits of Arase should be indicated by e.g. vertical lines; this is important because of statements like this: “The O⁺ density is quite low during most of this orbit” (Line 153). I am suggesting this for the sake of the reader – he should be able to locate the described feature relatively easily.

Added vertical lines and also labeled the orbits in panel D.

7. Line 23: “because the solar wind $\text{He}^{++}/\text{H}^+$ ratio can vary by as much as a factor of 10 during a storm” – please provide a comment/explanation and one or more references here.

Added “due to the enhanced heavy ion abundance associated with coronal mass ejections (CME’s)” and include two references. This is now in lines 111-113 (tracked changes version)

8. Lines 161-163: “Arase observes two ~ 1.5 -hr time periods when the $\text{He}^{++}/\text{H}^+$ ratio drops significantly below the solar wind value. These time periods are also accompanied by increases in O^+ flux and density” – O^+ flux and density seem to decrease rather than increase. Please provide comment/explanation.

I think this comment comes just from the difficulty in lining things up by eye in panels that are far apart. I have added green vertical lines around the time periods to guide the eye. The O^+ does increase while the $\text{He}^{++}/\text{H}^+$ decreases.

9. Lines 201-203: “The enhanced reconnection driven by this strongly southward field apparently triggers dynamic behavior on the nightside that drives the outflow” – please provide some suggestion or hypothesis on the “triggering of dynamic behavior” (both with regard to “triggering” and with regard to the “dynamic behavior”).

Changed to: The enhanced reconnection driven by this strongly southward field apparently triggers reconnection on the nightside that sends energy into the nightside auroral region, driving outflow. (lines 270-271, tracked changes version).

Reviewer #2 (Remarks to the Author):

This is a review of the paper “The source of the magnetotail plasma during a geomagnetic storm” submitted by Kistler et al. This paper looks at a fundamental question in magnetospheric science, namely what is the origin of near-Earth plasma. As the paper points out, there are two sources of magnetospheric plasma: the solar wind and the ionosphere. O^+ clearly comes from the ionosphere but H^+ can come from either source. This paper examines the source of plasma in the plasmashet using a combination of Arase, MMS, Cluster, WIND/SWE, and Cluster data. They also overcome the ambiguity regarding the source of H^+ by using simultaneous monitoring of the $\text{He}^{++}/\text{H}^+$ ratio in the solar wind and the magnetosphere. It has been shown in past work that this ratio could be used to determine how much of the H^+ is of ionospheric or solar wind origin (although simultaneous monitoring was not available in the past work). Throughout the course of an event, the authors successfully track the composition change and the origin of the plasma.

Overall, this is an excellent and well written study well deserving of publication. It addresses an issue of fundamental importance to magnetospheric science (the origin of near-Earth plasma). It is important for space weather as magnetospheric composition affects the decay of the ring current. This topic also relates to the issue of ionospheric outflow which has broad applicability to planetary science and atmospheric evolution. This study employs excellent data analysis to convincingly show how H^+ of ionospheric origin can dominate the storm main phase before the

O⁺ rich outflow arrives. As H⁺ has often been neglected as an ionospheric source of plasma in comparison to O⁺ of less ambiguous origin, this is a compelling result. These results are also consistent with some recent simulations as the authors point out. It is my recommendation that this work be published as is.

Thank you for your comments.

Reviewer #3 (Remarks to the Author):

By combining data from multiple spacecraft, the authors present a convincing story on the time evolution and origins of different ion species in the plasma sheet during a geomagnetic storm.

Some earlier studies have considered the ion composition in the plasma sheet during storms, but the novelty of this study is in the simultaneous data from different locations in the magnetosphere – in the near-Earth (Arase) and far (MMS) plasma sheet, as well as crossing the lobes and polar cap (Cluster), and also upstream in the solar wind (Wind). By comparing the data from different ion species at different locations, they are able to give a clear global description of the ion composition at different times during the storm. I found that the paper is well written, and the conclusions are well supported by the data presented. The data has been made available, so that others are able to reproduce the study.

Some minor suggestions –

1. There are some older references that are relevant to this work:

Gloeckler, G., & Hamilton, D. C. (1987). AMPTE ion composition results. *Physica Scripta*, 1987(T18), 73.

Shelley, E. G. (1986). Magnetospheric energetic ions from the Earth's ionosphere. *Advances in space research*, 6(3), 121-132.

These references have been added. (lines 102-104, tracked changes version)

2. Since there is a lot of free space in Fig. 1, I would suggest adding more labels for the different regions of the magnetosphere to help to make the article more accessible to a general audience.

Added labels for the major regions and boundaries.

REVIEWERS' COMMENTS

Reviewer #1 (Remarks to the Author):

The authors have successfully responded to my comments, especially with regard to the quantitative information instead of the qualitative discussion in several critical parts of the paper. Congratulations for this excellent study!

I have only four suggestions before publication:

1. “The enhanced reconnection driven by this strongly southward field apparently triggers reconnection on the nightside that sends energy into the nightside auroral region, driving outflow”: please provide some physics for the phrase “sends energy ... driving outflow”.

2. The topic of ionospheric outflow timing and ionospheric feeding of the magnetotail plasma sheet had been addressed by Daglis and Axford in 1996 and should be cited:

Daglis, I. A., and W. I. Axford, Fast ionospheric response to enhanced activity in geospace: Ion feeding of the inner magnetotail, *J. Geophys. Res.*, 101, 5047–5065, 1996.

3. The importance of plasma composition, particularly for intense storms, has been addressed by the study:

Daglis, I. A., Kozyra, J. U., Kamide, Y., Vassiliadis, D., Sharma, A. S., Liemohn, M. W., Gonzalez, W. D., Tsurutani, B. T., and Lu, G.: Intense space storms: Critical issues and open disputes, *J. Geophys. Res.-Space*, 108, 1208, <https://doi.org/10.1029/2002JA009722>, 2003.

which should be cited.

4. There is a typo in Line 214: “It finally increases at on Sept 8” - delete “at”

Response to the reviewer:

We thank the review for the helpful comments. We have addressed the final suggestions in the following way. All line numbers refer to the “tracked changes” version.

1. “The enhanced reconnection driven by this strongly southward field apparently triggers reconnection on the nightside that sends energy into the nightside auroral region, driving outflow”: please provide some physics for the phrase “sends energy ... driving outflow”.

Added “via Poynting flux and precipitating particles” (line 336)

2. The topic of ionospheric outflow timing and ionospheric feeding of the magnetotail plasma sheet had been addressed by Daglis and Axford in 1996 and should be cited: Daglis, I. A., and W. I. Axford, Fast ionospheric response to enhanced activity in geospace: Ion feeding of the inner magnetotail, *J. Geophys. Res.*, 101, 5047–5065, 1996.

Added this reference where I discuss the nightside auroral outflow (line 53)

3. The importance of plasma composition, particularly for intense storms, has been addressed by the study:
Daglis, I. A., Kozyra, J. U., Kamide, Y., Vassiliadis, D., Sharma, A. S., Liemohn, M. W., Gonzalez, W. D., Tsurutani, B. T., and Lu, G.: Intense space storms: Critical issues and open disputes, *J. Geophys. Res.-Space*, 108, 1208, <https://doi.org/10.1029/2002JA009722>, 2003.
which should be cited.

Added this to the list of references addressing the impacts of composition on storms. (line 61)

4. There is a typo in Line 214: “It finally increases at on Sept 8” - delete “at”
Fixed